# Deep Reinforcement Learning of Universal Policies with Diverse Environment Summaries

## Abstract

Deep reinforcement learning has enabled robots to complete complex tasks in simulation. However, the resulting policies do not transfer to real robots due to model errors in the simulator. One solution is to randomize the simulation environment, so that the resulting, trained policy achieves high performance in expectation over a variety of configurations that could represent the real-world. However, the distribution over simulator configurations must be carefully selected to represent the relevant dynamic modes of the system, as otherwise it can be unlikely to sample challenging configurations frequently enough. Moreover, the ideal distribution to improve the policy changes as the policy (un)learns to solve tasks in certain configurations. In this paper, we propose to use an inexpensive, kernel-based summarization method method that identifies configurations that lead to diverse behaviors. Since failure modes for the given task are naturally diverse, the policy trains on a mixture of representative and challenging configurations, which leads to more robust policies. In experiments, we show that the proposed method achieves the same performance as domain randomization in simple cases, but performs better when domain randomization does not lead to diverse dynamic modes.

## 1 Introduction

Reinforcement learning (RL, Sutton & Barto (1998)) is a powerful paradigm that has enabled impressive results in controlled environments such as games (Mnih et al., 2015) and simulated robotic systems (Lillicrap et al., 2015). However, it has proven difficult to transfer these successes to physical systems, as the simulations that are used for training are only an approximation of the real world. Consequently, the trained policies may fail to complete the task on the physical system and expensive and impractical retraining in the physical world might be required. In this paper, we consider the problem of training policies that are *robust* with respect to these errors in the simulator. Specifically, we present an algorithm that adaptively identifies a representative and *diverse* set of simulator configurations, which is then used to train the policy, see Fig. 1.

**Related work** Training policies that are robust towards model errors in robotics has mostly been considered in model-based RL. There, Bayesian models have been used in order to compute policies that are robust towards the worst-case model (Akametalu et al., 2014; Berkenkamp et al., 2017). By estimating and improving the model online, these methods allow for safe policy improvement on the real system. However, these methods suffer from the curse of dimensionality.

In high-dimensional state spaces, model-free RL has been the most successful (Kober & Peters, 2014). In particular, methods based on the natural policy gradient (Kakade, 2002), e.g., proximal policy optimization (PPO) (Schulman et al., 2017), and deterministic policy gradients (DPG) (Silver et al., 2014) together with neural network policies have been successful. These methods estimate the gradient of the performance based on trajectories induced by the current policy. However, the resulting policies are not robust towards model errors and may fail to work on the real system. One solution is to consider transfer learning (Taylor & Stone, 2009) or multi-task learning and environments. Wulfmeier et al. (2017) uses data gathered on the real system in order to improve the policy on the simulator in parts of the state space that were visited by the real system. An alternative approach by Marco et al. (2017) uses a Bayesian approach to transfer knowledge about the optimal policy from simulation to a robot. Both approaches require experiments on the real system. A related

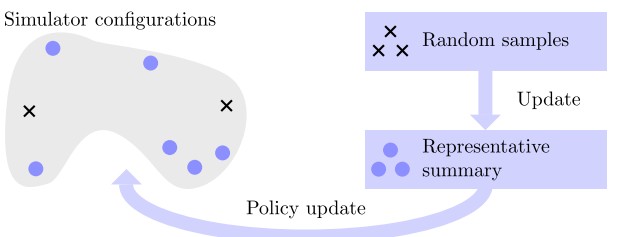

Figure 1. Illustration of the algorithm. In order to train robust and universal policies effectively, our algorithm maintains a representative set of environments for policy training. This set is diverse in state-trajectories and captures diverse dynamic modes of the underlying system. The summary is updated by evaluating random configurations, which leads to a data-efficient algorithm.

approach is used by (Rusu et al., 2017), which first trains a policy in simulation and then extended the neural network for additional training on the real system.

Alternatively, one can render policies more robust towards model errors on the simulator directly, without experiments on the physical system. This idea is similar to multi-task learning, where the goal is to train policies that perform well across multiple tasks (Devin et al., 2017; Teh et al., 2017). Training a policy accross multiple simulators has first been used in computer vision, where color and lighting are randomized to make the resulting policy transfer better to real-world conditions (Sadeghi & Levine, 2017; Tobin et al., 2017). Tan et al. (2018) use the same idea to randomize physical parameters of the robot. However, while lighting conditions affect the state observations directly and are designed to be diverse, sampling physical parameters uniformly does not necessarily lead to diverse state trajectories. We will show that this difference can cause randomization to perform poorly when difficult parameter combinations are only sampled infrequently. Instead of considering average performance, both Yu et al. (2017) and Rajeswaran et al. (2016) aim to optimize the worst-case performance across all configurations. To this end, they sample simulator configurations and only use the ones with the worst performance under the current policy for training. However, these approaches require a large number of simulations, most of which are not used to update the policy, and are thus computationally expensive. Another issue that can arise in these settings is catastrophic forgetting (Goodfellow et al., 2013), where by training only on the worst-case simulator the policy unlearns how to perform in previously solved simulator configurations. Another approach by Pinto et al. (2017) considers adversarial training, where an adversary locally disrupts the simulation. However, local approaches typically do not lead to policies that perform well across all configurations.

We avoid these problems by using a small but representative summary of simulator configurations instead. These kind of representative summaries have previously been considered for sensor placement (Krause et al., 2008), where the goal is to cover a space with sensors in order to increase the amount of information gained while minimizing the cost of placing the sensors. Such representative summaries have also been used in sequential action selection (Dey et al., 2013), path planning (Dey et al., 2012), and information gathering tasks (Choudhury et al., 2017; Binney et al., 2010). These works exploit structure in the problem known as submodularity (Nemhauser et al., 1978), which leads to tractable algorithms with provably close-to-optimal performance.

**Contribution** In this paper, we use ideas from data summarization in order to obtain a representative summary of simulator configurations that capture the diverse and relevant dynamic modes of the underlying system. Our key contributions are, first, to propose a similarity metric that allows us to quantify differences in simulator configurations that are invariant to parameter transformations by encoding dynamic behavior that is physically meaningful. Specifically, we use a kernel that measures similarity of simulator configurations based on the trajectories induced by the current policy. Secondly, we propose a tractable algorithm based on streaming submodular optimization to efficiently and adaptively compute a summary of configurations based on the similarity metric. Together, these two contributions ensure that the resulting summary covers the space of state-trajectories induced by different simulator configurations well. Thus, by training on this diverse summary, the resulting policies perform well across all dynamic modes of the simulators. In cases where uniform sampling of configurations already leads to diverse dynamic modes, our method achieves the same performance and robustness as domain randomization. However, when this is not the case and domain knowledge would be needed to specify a suitable sampling distribution, training on diverse configurations leads to significantly improved performance and robustness.

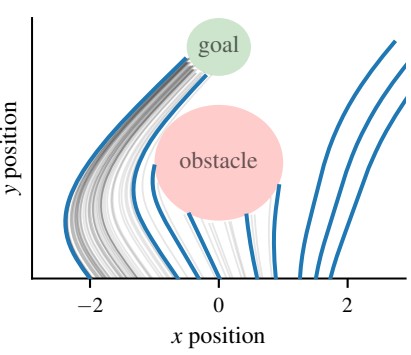

(a) Example trajectories and summary.

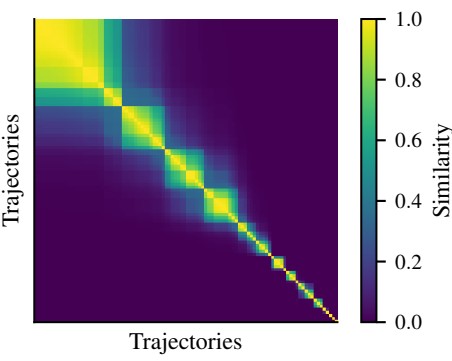

(b) Correlations between environments.

Figure 2. The task is to reach the goal without crashing into the obstacle, Fig. 2a. The simulator is parameterized by the transformed initial position $m \in \mathcal{M}$, so that uniform sampling leads to more trajectories on the left of the obstacle (gray). Thus, trajectories on the right are underrepresented during training. Instead, our algorithm analyzes the similarity between trajectories in Fig. 2b and computes a diverse and representative summary (blue lines), which counteracts the sampling bias.

## 2 PROBLEM STATEMENT AND BACKGROUND

We consider a simulator with state space $\mathcal{S} \subseteq \mathcal{R}^d$ and actions in $\mathcal{A}$. During simulation, the whole state space is observed. The simulator is parameterized by a set of physically meaningful parameters $\mathcal{M}$. These parameters represent different simulator configurations by, for example, modify the task or changing physical properties of a robot, such as mass or friction. The goal is to train a single policy that performs well across all the possible configurations in $\mathcal{M}$. Since this set represents all uncertainties about the real-world system, policies that perform well across all configurations are likely to perform well in the real-world too. The simulator parameters affect the transition dynamics, so that each choice of parameters $m \in \mathcal{M}$ encodes a separate Markov decision process $\langle \mathcal{S}, \mathcal{A}, \mathcal{T}_m, c, \gamma \rangle$, where the stochastic transition dynamics $\mathcal{T}_m$ depend on the simulator parameters $m$. Performance for a fixed simulator configuration $m \in \mathcal{M}$ under a policy $\pi_\theta$ is measured via the expected $\gamma$-discounted future rewards $r(\cdot)$, which is given by $J_\theta(m) = \mathbb{E}_{a_t \sim \pi_\theta(s_t), s_{t+1} \sim \mathcal{T}_m(s_t, a_t)}[\sum_{t=0}^{\infty} \gamma^t r(s_t, a_t)]$. This objective can be maximized for a fixed model $m$ by estimating the gradients with respect to the policy parameters $\theta$ from state trajectories obtained through simulated rollouts (Kakade, 2002).

To achieve robustness across all configurations $\mathcal{M}$, ideally we would optimize the worst-case performance directly: $\max_\theta \min_{m \in \mathcal{M}} J_\theta(m)$. However, this problem is intractable and, even if it was, gradient-based updates based on the worst-case configuration may not converge for neural network policies. Consider the problem in Fig. 2a, where the goal is to reach a target behind an obstacle and the simulator is parameterized by the starting position at the bottom. If the initial worst-case starting position is left of the obstacle, the gradient update will push state trajectories away from the obstacle towards the right. In contrast, when the starting position is right of the obstacle, the policy update will push the trajectory to the left. These two opposing updates can lead to an oscillating cycle during the optimization with slow or no convergence to the optimal policy. The underlying problem is that the gradient is estimated based on trajectories that do not cover the state-space sufficiently, so that policy improvement to the left of the obstacle leads to worse performance on the right side and vice-versa. This kind of forgetting is well-known in multi-task reinforcement learning (Goodfellow et al., 2013) and was also hinted at in (Rajeswaran et al., 2016).

To avoid this issue in practice, one can instead replace the minimum by a finite sum over a small, but representative set of models $\mathcal{M}_s \subset \mathcal{M}$,

$$\max_\theta \sum_{m \in \mathcal{M}_s} J_\theta(m), \tag{1}$$

which can be easily solved using standard batch policy gradient methods. The choice of $\mathcal{M}_s$ is critical, since it needs to capture all relevant dynamic modes in $\mathcal{M}$ that cover the state space in order to avoid training issues as in Fig. 2a. Domain randomization typically samples $\mathcal{M}_s$ uniformly at

random from $\mathcal{M}$ after every policy update. However, this only ensures coverage in the parameter space $\mathcal{M}$. The resulting coverage in terms of state-trajectories depends on the parameterization of $\mathcal{M}$, transition dynamics, and the policy. For example, in the model in Fig. 2a uniform sampling in parameter space leads to trajectories that pass to the left of the obstacle more frequently than to the right. This leads to a training bias, where parameters on the right have worse performance, since the corresponding trajectories are underrepresented during training. Instead, the set $\mathcal{M}_s$ should be selected to be diverse in state-trajectories, as shown by the blue lines in Fig. 2a. This ensures that also tasks that parameter configurations that are sampled infrequently under the uniform distribution achieve good performance. As a consequence, the resulting policies trained on diverse trajectories are more robust.

In summary, we want to select a set $\mathcal{M}_s$ that represents diverse dynamic modes and state-trajectories of the underlying system, without exploiting domain knowledge about how to parameterize $\mathcal{M}$. Moreover, we want to compute this set efficiently, without requiring large amount wasteful rollouts that are not used for training as in (Yu et al., 2017; Rajeswaran et al., 2016).

**Submodular Optimization** As stated previously, the parameters in $\mathcal{M}_s$ should represent diverse induced state-trajectories. This kind of diversity requirement has previously been considered, especially in form of the sensor placement problem (Krause et al., 2008). There, the goal is to cover a space with sensors in order to maximize the amount of information gained. In general, this is a difficult combinatorial optimization problem, which requires evaluating every possible combination of sensors in order to find the best one. To obtain a tractable and probably close-to-optimal algorithm, Krause et al. (2008) exploit a diminishing returns property of the problem known as submodularity.

Conceptually, submodularity states that adding a sensor to a small set of existing sensors always provides more information than when we already have a large set of sensors that cover the space. More precisely, a set function $f$ that maps a subset $\mathcal{M}_s$ of the domain $\mathcal{M}$ to the reals is called submodular if, for any two summaries $\mathcal{M}_s \subset \mathcal{U} \subset \mathcal{M}$ and any additional new element $m \in \mathcal{M}$, it holds that that $f(\mathcal{M}_s \cup \{m\}) - f(\mathcal{M}_s) \geq f(\mathcal{U} \cup \{m\}) - f(\mathcal{U})$; that is, for the larger set $\mathcal{U}$ adding a new sensor $m$ helps less than for the small set $S$. Another natural assumption is monotonicity, which means that larger summaries are always better, $f(\mathcal{U}) \geq f(\mathcal{M}_s)$. Maximizing monotone submodular functions can be done efficiently using the greedy algorithm with an approximation ratio of at least $1 - 1/e \approx 0.68$ relative to the optimal solution (Nemhauser et al., 1978). The greedy algorithm starts with the empty set $\mathcal{M}_s = \{\}$ and greedily selects the element that maximally improves the function value within one step, until a constraint on the size of $\mathcal{M}_s$ is reached.

$$\mathcal{M}_s \leftarrow \mathcal{M}_s \cup \arg\max_{m \in \mathcal{M}} f(\mathcal{M}_s \cup \{m\}). \tag{2}$$

**Streaming Submodular Optimization** The greedy algorithm in (2) iteratively constructs the set $\mathcal{M}_s$ by evaluating all possible elements $m$ in $\mathcal{M}$ that could be added to the current set. This requirement is relaxed by streaming submodular optimization algorithms, which assume that the data comes in as a stream and one has to decide for each element whether to add it to the summary or disregard it. The STREAM-GREEDY algorithm by Krause & Gomes (2010) considers a conceptually straight-forward extension of the greedy algorithm (2). For each new element $m$ in the stream, they consider if there is a benefit of replacing one of the current elements $m' \in \mathcal{M}_s$ in the current summary with $m$,

$$(m', m) = \arg\max_{m' \in \mathcal{M}_s, m \in \mathcal{M}_s \cup \{m\}} f(\mathcal{M}_s \setminus \{m'\} \cup \{m\}), \quad \mathcal{M}_s \leftarrow \mathcal{M}_s \setminus \{m'\} \cup \{m\}, \tag{3}$$

This algorithm also enjoys constant-factor optimization guarantees (Krause & Gomes, 2010). The bounds can be further improved with a more complicated algorithm (Badanidiyuru et al., 2014).

## 3 MAIN ALGORITHM

In this section, we show how to use techniques from streaming submodular optimization to obtain representative summaries of simulator configurations that can be used to train robust neural network policies. First, we need to specify an objective set function $f(\cdot)$ that measures the quality of a given summary $\mathcal{M}_s$. Following the discussion in Sec. 2, we aim to find diverse dynamics modes, which is similar to the sensor placement problem in Krause et al. (2008). There, the objective is

$$f(\mathcal{M}_s) = \sum_{\mathcal{M}_s} \log \left| I + \sigma^{-2} K_{\mathcal{M}_s} \right|, \tag{4}$$

which computes a diversity score based on a kernel matrix $K_{\mathcal{M}_s} \in \mathbb{R}^{N \times N}$, where $N$ is the size of the summary. The kernel matrix is defined through a kernel function, $[K_{\mathcal{M}_s}]_{(i,j)} = k(m_i, m_j)$ that measures the similarity of two elements in the summary set $\mathcal{M}_s = \{m_i\}_{i=0}^N$. We provide more details of how to select the kernel later. The diversity measure in (4) has an information-theoretic interpretation as the mutual information, if we obtain noisy measurements at parameters in $\mathcal{M}_s$ of a Gaussian distribution with covariance $K_{\mathcal{M}_s}$. Intuitively, (4) assigns high values to summaries that are very diverse, as measured under the kernel. The parameter $\sigma^{-2}$ is a tuning parameter that has no larger importance to us. We set $\sigma^{-2} = 1e8$. See Krause et al. (2008) for more details.

**Kernel Selection for Diversity** The choice of kernel is critical, since it encodes our notion of coverage. From the motivation in Sec. 2, it follows that we want to cover the state space $\mathcal{S}$ in terms of trajectories that cover the state-space. Typically, kernels are defined on the set of parameters $m \in \mathcal{M}$ directly. This would lead to behavior similar to domain randomization, as it only considers coverage in parameter space. However, as seen in Fig. 2a close to the obstacle, a small change in parameters can lead to drastically different trajectories. This violates the assumptions made by typically used kernels. As a result, coverage in terms of simulator parameters can perform poorly, an intuition that is also confirmed by our experiments in Sec. 4.

To avoid this issue, we propose to define the kernel directly on the trajectories induced by simulator parameters. That is, for two simulator configurations $m, m' \in \mathcal{M}$, we evaluate the state trajectories $\tau_m$ and $\tau_{m'}$ by rolling out under the current policy $\pi_\theta$. It is possible to define several measures of similarity based on these trajectories. We choose to measure similarity based on the exponentiated $\ell_1$-distance, which is called the laplace kernel,

$$k(m, m') = \exp\left(-\sum_{i=1}^{\dim(\mathcal{S})} \frac{1}{l_1} \|\tau_m^i - \tau_{m'}^i\|_1\right), \tag{5}$$

where $\tau_m^i$ is the $i$th state-component of the state trajectory $\tau_m$ and $l_i$ is a scaling factor. This kernel considers two trajectories to be similar if their weighted difference in states for each time step is small under the 1-norm. This error is physically meaningful, since it is expressed in terms of states. For example, it might correspond to the average difference in positions of two robots. The scalars $l_i$ determine the relative weighting of state dimensions. Trajectories with state difference smaller than $l_i$ are considered to very similar, with an exponential decrease in similarity as the error increases. If the two trajectories are of different lengths, we hold the last state of the shorter trajectory. In terms of the parameter space $\mathcal{M}$, this kernel often encodes non-stationary phenomena, where a small change in parameters can sometimes lead to a large change in similarity. For example, the kernel is used to compute similarities in Fig. 2b, which leads to the summary (blue lines) in Fig. 2a. While other norms and kernels can be used, e.g., (Wilson et al., 2014), we have found that this kernel is easy to tune and specify using readily available intuition about the system at hand.

**Optimization Algorithm** With the kernel defined, we now consider the optimization problem

$$\max_{\mathcal{M}_s \subset \mathcal{M}, |\mathcal{M}| < N} f(\mathcal{M}_s), \tag{6}$$

where the goal is to find a summary $\mathcal{M}_s$ of size $N$ that maximizes the objective (4). The objective function (4) is monotone submodular (Krause & Guestrin, 2005), which means it can be efficiently optimized using the greedy algorithm. However, unlike in the classic submodular optimization case, we face two additional challenges. Firstly, the classic submodular optimization setting assumes that it is possible to compute the kernel matrix over all possible, discrete parameters in $\mathcal{M}$. However, in our setting the set $\mathcal{M}$ is continuous and evaluating the kernel (similarity) requires a complete rollout of the system under the current policy. Since this is intractable, it is impossible to use the greedy algorithm in (2) in order to solve (6). Secondly, every time we update the parameters $\theta$ of the policy $\pi_\theta$, the system trajectories, and thus also the objective function (4), change. This is in contrast to submodular optimization, which considers the objective function $f$ to be fixed.

We overcome these issues by using an algorithm inspired by streaming submodular optimization, see Sec. 2. The resulting algorithm is summarized in Algorithm 1. It starts by sampling a summary set $\mathcal{M}_s$ uniformly at random from the domain $\mathcal{M}$ in Line 2. Then the algorithm proceeds to the policy optimization loop. At each iteration, it starts by rolling out trajectories for each configuration $m \in \mathcal{M}_s$ under the current policy in the simulator. This updates the objective function (4) based on the new trajectories under the policy. As a next step, the algorithm samples $N_s$ new simulator configurations uniformly at random as in standard domain randomization, and updates the summary $\mathcal{M}_s$

---

**Algorithm 1** Diverse Domain Summarization

---

1: **Input:** $\pi_\theta, \mathcal{M}, N_s$
2: Sample summary $\mathcal{M}_s \subset \mathcal{M}$ uniformly at random.
3: **for** $n = 1, \ldots$ **do**
4:      Evaluate trajectories $\tau_m$ for simulators $m \in \mathcal{M}_s$.
5:      **for** $k = 1, \ldots, N_s$ **do**
6:          Sample parameter $m \in \mathcal{M}$ uniformly at random.
7:          Evaluate trajectory $\tau_m$.
8:          Update summary $\mathcal{M}_s$ with (3) and objective (4).
9:      Policy update based on trajectories $\tau_m, m \in \mathcal{M}_s$.

---

by replacing elements in $\mathcal{M}_s$ whenever an improvement in the objective is possible in Lines 6-8. Lastly, the algorithm uses the current summary $\mathcal{M}_s$ and already computed, corresponding trajectories in order to update the policy using standard policy gradient methods. While the algorithm still samples and evaluates $N_s$ simulator configurations uniformly at random, this number can be small, as the policy, and therefore the optimal summary with respect to $f$, change relatively slowly between policy updates.

When the model set $\mathcal{M}$ is discrete, we can recover theoretical guarantees for the solution quality of the STREAM-GREEDY algorithm (3) for solving (6). The only additional requirement is that the objective function changes slowly enough in terms of maximum pointwise distance, which is typically satisfied if the learning rate of the policy update is chosen small and the dynamics, policy, and rewards are Lipschitz continuous. In this case, it follows from the theoretical analysis in (Krause & Gomes, 2010, Thm. 4) that this variant of the algorithm can track the optimal summary up to a constant factor.

**Practical considerations** The main tuning parameters of Algorithm 1 are the diversity measure and kernel, the size of the summary $N$, and the number of simulator configurations that are sampled at each iteration $N_s$. As discussed, the kernel has a physically meaningful interpretation in terms of distance of trajectories. However, while the relative weighting of the parameters $l$ is physically meaningful, the overall magnitude is arbitrary. Overall we have found that multplying all $l_i$ by a constant does not affect the resulting summary significantly. Only the two limit cases, where $l_i$ is so large that almost all summaries are equivalent or where $l_i$ is so small that the difference in similarity drops below numerical accuracy, can cause problems. We avoid these by making sure that the current solution $f(\mathcal{M})$ lies between 20% and 70% of the maximal possible value $f$ that would be achieved if $K_{\mathcal{M}_s}$ was the identity matrix. We enforce this constraint via an efficient line search. The appropriate size of the summary depends on the complexity of the dynamics, the number of different dynamic modes one expects to cover, and on how many rollouts are required to obtain sufficiently low variance in the policy gradients. The number of random rollouts should be as large as possible, but we have found that $N_s = 0.2N$ is typically sufficient.

Many variants of Algorithm 1 are possible, which may combine random samples into the gradient-based updates or consider additional weightings inside the summary more akin to the algorithms in Yu et al. (2017); Rajeswaran et al. (2016). We consider this paper a first step towards highlighting the importance of coverage in terms of state-trajectories when training universal policies.

## 4 EXPERIMENTS

In this section, we evaluate Algorithm 1 on two benchmark problems: The illustrative example in Fig. 2a using DPG, and a high-dimensional robotic environment example based on PyBullet (Coumans & Bai, 2016) using PPO. As performance measures, we consider the average performance over $\mathcal{M}$ and the diverse performance that is computed by sampling large amounts of environments and using the greedy algorithm to compute a representative summary. Note that this is computationally expensive and that Algorithm 1 only has access to small batches for training, which means its summary can only ever approximate the diverse performance objective during training.

**Illustrative Example** We consider the illustrative example in Fig. 2a. The dynamics are given by the Dubins car model; a simplistic model of a car that moves at constant velocity. It has two states that

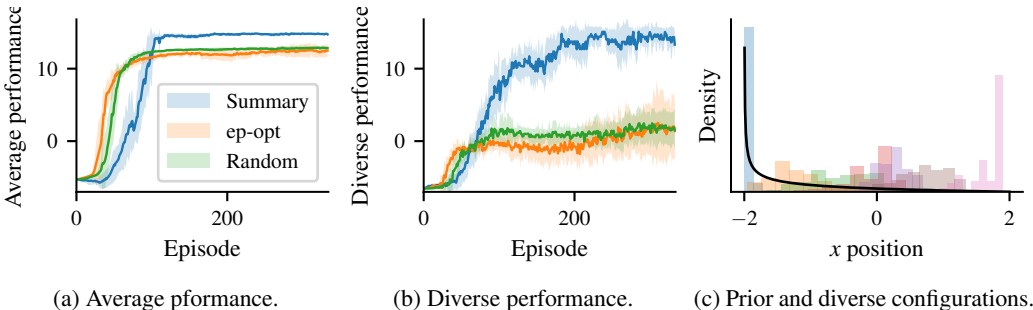

(a) Average pformance.  (b) Diverse performance.  (c) Prior and diverse configurations.

Figure 3. Example run of Algorithm 1 for the example system in Fig. 2a. Random sampling in the parameter space leads to initial conditions that are biased to the left of the obstacle (black line, Fig. 3c). While the average performance in Fig. 3a is good for all methods, only our method is able to counteract the sampling bias by keeping track of diverse environments in Fig. 3c. This is also reflected in the performance over a diverse summary of environments in Fig. 3b. EP-opt only works with significantly more samples than available here.

correspond to the $x$ and $y$ position. With the steering angle $a \in [-\pi/4, \pi/4]$ as control input, the continuous time dynamics are given by $\dot{x} = \sin(a)$ and $\dot{y} = \cos(a)$. The simulator is parameterized by the initial $x$-position, $\mathcal{M} = [-2, 2]$, while the initial $y$-position is equal to $-2$. An episode terminates with a reward of $+10$ if the car enters the goal circle with radius $0.5$ at $(0, 2)$ and a reward of $-10$ if it enters the obstacle ball at the origin with radius $1$, or if the state trajectory diverges and leaves the ball with radius $3$. Intermediate rewards are equal to $r(s, a) = 0.2 - 0.1\|s - (0, 2)\|_2^2$, which consists of a $0.2$ bonus for staying alive and a cost that penalizes distance from the goal.

This is an easy RL problem. To make it more challenging, we reparameterize the initial position so that uniform sampling from $\mathcal{M}$ leads to initial $x$ positions that follows a beta distribution, $x_0 \sim B(0.5, 2)$. This causes typical trajectories to lie on the left of the obstacle ($x < 0$), see Fig. 2a (gray lines). This is representative of higher-dimensional problems, where one may see only few random samples for certain, challenging situations. As a result, the average error over parameters $m$ is no longer a suitable objective. To overcome this challenge, we use Algorithm 1 with a summary size of $N = 7$ in order to cover diverse dynamics modes, similar to the blue lines in Fig. 2a. The kernel is defined as in (5) and considers the average distance with $l_0 = l_1$. As a result, the algorithm aims to find parameters that lead to different trajectories (e.g., starting from the left and right).

We use the DDPG algorithm from (Lillicrap et al., 2015) to train a deterministic policy. The only difference is that the value function baseline must additionally depend on the environment parameter $m$. This is necessary, since DDPG is an actor-critic algorithm and the value function must be able to capture the different future values depending on the parameters. The policy is a two-layer neural network with relu activations and a tanh activation at the output to saturate actions. It only depends on the state. For each environment that is selected by the algorithm, we rollout in simulation with the policy corrupted by noise for 1000 time steps in order obtain policy gradient estimates.

We compare against two baselines. The first on is random sampling, which samples parameters uniformly and thus initial $x$-positions from the beta distribution. The second baseline is EP-OPT, the method from (Yu et al., 2017) and (Rajeswaran et al., 2016), which also samples uniformly, but where we only train on the worst 7 environments. For a fair comparison, each algorithm is allowed to select at most $N + N_s = 10$ rollouts and all algorithms use the same policy gradient method with the same tuning parameters and network structure.

The results can be seen in Fig. 3, which shows the average cumulative reward over initial positions $x$. Note that this is different from the model parameters $m$. Confidence intervals (shaded region) show the standard deviation across 10 different random seeds. We do not optimize over random seeds. The two baselines that sample randomly from the domain $\mathcal{M}$ solve the task with respect to their own metrics, which effectively considers initial conditions samples from the prior distribution in Fig. 3c (black line), and achieve high average performance in Fig. 3a. However, due to the sampling bias they only rarely encounter initial conditions on the right side of the obstacle, so that the resulting policy only works for starting positions to the left of the obstacle. This is reflected in the poor diverse

| Model | Average performance | Diverse performance |
|---|---|---|
| Summary | $2078.32 \pm 345.18$ | $1956.06 \pm 357.59$ |
| Random | $2056.40 \pm 575.05$ | $1845.58 \pm 750.19$ |

Table 1. Average and diverse performance on the PyBullet HalfCheetah environment across 8 random seeds.

performance in Fig. 3b. Under the constraint on the number of rollouts, EP-OPT does not work, since it is unable to conduct large amounts of rollouts to identify the worst environments. In contrast, Algorithm 1 does not resample at every step, but keeps track of a diverse summary. This summary covers all starting positions, as can be seen in Fig. 3c. Thus, the algorithm counteracts the sampling bias and allows it to solve all environments. This is also reflected in a high performance over diverse environments, even though the algorithm does *not* waste large amounts of samples to compute a diverse summary at every step. Notably, the summarization method achieves an even better average performance, since it can additionally solve the rare samples to the right of the obstacle. All methods solve the task successfully without the reparameterization.

**Robotics** We also apply our method to a higher-dimensional system. We use the simulated robotic half-cheetah provided by the PyBullet simulation environments (Coumans & Bai, 2016), but modify the simulation parameters. In particular, we introduce uncertainty in all seven body masses and the friction constant between the two feet and the ground, with errors of up to $50\%$ of the original parameters. We use the PPO implementation in Tensorflow Agents (Hafner et al., 2017) in order to train a two-layer relu network as a policy. The hyperparameters of the training algorithm are set as suggested in (Coumans & Bai, 2016; Hafner et al., 2017).

As in the previous example, we randomly sample parameters from the high-dimensional parameter set $\mathcal{M}$ and use the kernel from (5) to summarize this set. Since the states have different magnitudes, we normalize the position and rotation parameters in order to compare unit-free states in the kernel computation. To summarize the parameter space, we use $N = 24$ environment summaries with $N_s = 6$ random samples. The summary size is larger, since the higher-dimensional system has more diverse dynamic modes than the previous illustrative example. For ease of implementation and since, in higher dimensional system, the variance of the policy gradients becomes a significant factor, we train the robot on both the environment summaries and the $N_s$ random rollouts. This means that the resulting objective is a mixture of average and diverse performance. We compute the average and diverse performance of the resulting policies by sampling 1000 parameter configurations and rolling them out on the robot. The diverse performance is computed by using the greedy algorithm on this large set. This data information is not available to the training algorithm.

We train each policy for 250 million timesteps based on 8 random seeds. We did not optimize over the seeds. The resulting performances are shown in table Sec. 4. Both the method based on domain randomization and the summary achieve similar average performance, but the standard deviation of the summary method is significantly lower. Based on the insights from the previous illustrative example, a likely cause is that diverse summaries more regularly encounter difficult training scenarios than pure random sampling, which leads to more robust optimization behavior. The diverse performance of the summary method is both higher in expectation and has a standard deviation that is $50\%$ smaller than that of domain randomization. This shows that the summary has more consistent performance across all configurations, rather than focusing on the ones that are likely to be sampled.

In summary, it can be seen that the summarization method can help identify relevant dynamic modes and help improve the performance of the model across all state trajectories, independently of how likely they are to be sampled under domain randomization. Importantly, adding this randomization typically does not decrease average performance. However, it can help improve performance in situations that are not represented sufficiently during training.

## 5 CONCLUSION

We presented Algorithm 1, a data-efficient summarization method that maintains representative summaries of simulator configurations in order to train robust policies using policy gradient methods. We showed that these summaries are invariant under reparameterization and can help in challenging situations where uniform sampling might not be appropriate.

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
