# OpenReview forum: "Deep Reinforcement Learning of Universal Policies with Diverse Environment Summaries"
_ICLR.cc/2019/Conference_

### Official Review · AnonReviewer3 · 2018-10-28
**Empirical results are not convincing enough**

**Rating:** 5
**Confidence:** 4

**Review:**

This paper studies the problem of robust policy optimization, motivated by the fact that policies that work in simulations do not transfer well to real world. The authors propose to use the diversity measure on the roll out trajectories to select a diverse set of simulator configurations and train policies that can work well for all of those selected configurations.

Overall, I think the idea is interesting but it is not entirely clear why adding diverse configurations should result in good performance, and the experiments are very limited and not convincing enough.

Pros:
- The paper is easy to follow.
- The idea of using a diverse summary to do robust policy optimization is interesting.
- The diversity measure on the trajectories instead of the space of configuration parameters also intuitively makes sense since it takes into account that the similarity between two configuration parameters does not typically mean the similarity between their corresponding policies.

Cons:
- The setting of this paper seems to only work for the fully observable case with state space being in R^d, deterministic dynamics and deterministic policy (otherwise the diversity measure would be stochastic?). It would be good to clarify these in Sec. 2.
- For the example in Fig 2 and the first experiment, what I don't understand is why the initial state is not part of the policy.
- It is not clear if the reason that EP-OPT performed worse than the proposed approach is only because there are not enough rollouts for EP-OPT. This could be an unfair comparison.
- It would be good to show the comparison to EP-OPT for the second experiment as well.
- Two experiments may not be enough to verify valid performance since there could be a lot of randomness in the results.
- In page 6, it would be good to clarify that the summary being optimal is only with respect to f(M_s), but not the original problem of finding optimal policy.

---

### Official Review · AnonReviewer2 · 2018-11-03
**Efficient robust optimization in simulation through careful selection of diverse set of perturbations**

**Rating:** 6
**Confidence:** 4

**Review:**

This paper addresses the problem of finding a policy that will perform well in a real environment when training in a simulator that may have errors.  It takes the now standard approach of trying to find a policy that performs well in an ensemble of simulated environments that are perturbations of the basic simulator.   The question is:  how can we construct an ensemble that represents the uncertainty about the real world well while being small enough for computational efficiency.  The idea is to construct a diverse set of samples that represents the whole space of important variations in the simulation;  the particular novelty here is to ensure that the sample set attains coverage over the *behaviors* of the simulator rather than the parameters of the simulation.  This problem is made difficult by the fact that there is no finite space of samples to choose from and the fact that we don't have a natural distance metric on the simulator behavior.

The main positive contributions of the paper are:
- The view of the problem of selecting from an infinite set as one of streaming sub-modular optimization.  This is a nice idea that is new to me and seemed appropriate for the problem.
- The idea that we want diversity in behavior, and then the technical approach of defining a kernel on simulator parameter sets that depends on the trajectories that those parameters induce.

I do have a set of questions and concerns:
- Might it not be better (more robust) to use not just trajectories from the current policy, but from other policies as well, to compute the kernel on parameter sets?
- How do you get the length-scale parameters for the kernel?
- The confidence intervals in table 1 are too big to really justify firm conclusions;  it would be better to run the algorithms several more times, until the intervals pull apart.
- You say: "For ease of implementation  and since, in higher dimensional system, the variance of the policy gradients becomes  a significant factor, we train the  robot  on both the environment summaries and the N_s random rollouts."   This seems like it might be an important point that should be addressed earlier.  And, why does this ease implementation?
- I didn't understand:  "Diverse summaries are more consistent than pure random sampling."  What do you mean by consistent here?
- The metrics used in the empirical comparisons don't seem exactly right to me.  The goal of this work is to learn a policy that is robust in some sense (so  that,  e.g., it will do sim-2-real well).   We  really want  it to  work well in all  possible cases, not just in expectation or according to the sampling distribution you create, (I guess---since the paper said  the minimax criterion was desirable but difficult to work with).  So, then, it seems like  the best performance criterion would be to sample a whole lot of  domain parameters and report performance  on the worst  (rather than reporting performance on a distribution that's like the one you  trained it on or on an easy random one).

Overall, my view is that the idea is good, but somewhat small, and it hasn't really yet been proven to make a big difference.

---

### Official Review · AnonReviewer1 · 2018-11-04
**The proposed idea is not properly supported and is not convincing.**

**Rating:** 4
**Confidence:** 5

**Review:**

This paper presents a new method for learning diverse policies that can potentially transfer better to new environments. The proposed method aims to find simulation configurations that lead to diverse behaviors using “submodular optimizations” technique; an idea stemmed from past data summarization methods.

Pros:

-The paper deals with an important problem in RL which is learning robust policies that can transfer better

- Simple idea based on prior information theory literature is proposed.

- Good incorporation of past methods for improving robustness in RL.

Cons:

>>The paper has provided weak evidence and to support the effectiveness and significance of the proposed approach. The current analysis and experimental evaluations are not convincing.

Relevant baselines are missed and limited tasks are explored:
- A relevant prior work of Pinto et al. (2017) is not considered in the baselines for comparison.
- Only two tasks are considered in experiments which are not representative of the effectiveness of the proposed approach and does not provide convincing evidence that the proposed method performs better than prior works.
-In the HalfCheetah task only the performance of random baseline is shown and comparison with EpOpt is not reported
-Results of the base DDPG policy is reported in the experiments (Fig 3 and Table 1). For a thorough comparison these missing results are necessary.

>>Correct citation to the prior work of “domain randomization” is necessary:
- The second paragraph of related work section (first paragraph of page 2) explains that the idea of domain randomization has been first used in computer vision by Tobin et al. 2017. The idea of domain randomization was first proposed and deployed on a real robot platform in Sadeghi and Levine 2016 (Sadeghi, Fereshteh, and Sergey Levine. "CAD2RL: Real single-image flight without a single real image." arXiv preprint arXiv:1611.04201 (2016).) and was later used in Tobin et al. 2017 (this is also explained in the Tobin et al. 2017). Adding proper citation to Sadeghi and Levine 2016 is necessary.

>> Relevant prior works are missed:
- The proposed idea in the paper is relevant to multi-task RL (e.g. Teh Y, Bapst V, Czarnecki WM, Quan J, Kirkpatrick J, Hadsell R, Heess N, Pascanu R. Distral: Robust multitask reinforcement learning. InAdvances in Neural Information Processing Systems 2017 (pp. 4496-4506). Adding related discussion about prior multi-task RL methods is highly recommended for improving the paper.
- Citating several relevant prior RL methods that deal with transfer learning is missed. A few examples are:

Rusu AA, Vecerik M, Rothörl T, Heess N, Pascanu R, Hadsell R. Sim-to-real robot learning from pixels with progressive nets. arXiv preprint arXiv:1610.04286. 2016 Oct 13.

Rusu AA, Rabinowitz NC, Desjardins G, Soyer H, Kirkpatrick J, Kavukcuoglu K, Pascanu R, Hadsell R. Progressive neural networks. arXiv preprint arXiv:1606.04671. 2016 Jun 15.

Rusu AA, Colmenarejo SG, Gulcehre C, Desjardins G, Kirkpatrick J, Pascanu R, Mnih V, Kavukcuoglu K, Hadsell R. Policy distillation. arXiv preprint arXiv:1511.06295. 2015 Nov 19.


>> The format of the paper needs to be improved.
- The introduction section is rather incomplete and does not properly motivate the problem and the proposed solution.
-The organization of the experimental section should be improved. It is hard to follow the experiments and find the key experimental results in the current version of the paper.

---

### Meta-Review · Area_Chair1 · 2018-12-14

**Confidence:** 4
**Recommendation:** Reject

**Metareview:**

The paper proposes an approach to learn policies that can effectively transfer to new environments. The perspective on this problem from the perspective of streaming submodular optimization is nice; the paper introduces new ideas that are likely of interest to the ICLR community. Unfortunately, there are significant concerns about how convincing the results are. Multiple reviewers were concerned about there only being two experiments, and the lack of comparison to ep-opt on the half-cheetah experiment. Without a more solid empirical validation of the ideas, the paper does not meet the bar for publication at ICLR.